# The Problem of Host and Pathogen Genetic Variability for Developing Strategies of Universally Efficacious Vaccination against and Personalised Immunotherapy of Tuberculosis: Potential Solutions?

**DOI:** 10.3390/ijms24031887

**Published:** 2023-01-18

**Authors:** Peter A. Bretscher

**Affiliations:** Department of Biochemistry, Microbiology and Immunology, Health Sciences Building, 107 Wiggins Road, Saskatoon, SK S7N 5E5, Canada; peter.bretscher@usask.ca

**Keywords:** tuberculosis, mycobacteria and host gene expression, lung inflammation, TB latency, TB vaccination, TB immunotherapy

## Abstract

Rational vaccination against and immunotherapy of any infectious disease requires knowledge of how protective and non-protective immune responses differ, and how immune responses are regulated, so their nature can be controlled. Strong Th1 responses are likely protective against *M tuberculosis.* Understanding how immune class regulation is achieved is pertinent to both vaccination and treatment. I argue that variables of infection, other than PAMPs, primarily determine the class of immunity generated. The alternative, non-PAMP framework I favour, allows me to propose strategies to achieve efficacious vaccination, transcending host and pathogen genetic variability, to prevent tuberculosis, and personalised protocols to treat disease.

## 1. The Impact of Our Other Research on Our Research on Tuberculosis

I have been interested in the immunology of tuberculosis for several decades. The research of my students and myself in related areas had a considerable impact on our thinking about tuberculosis. I initially immersed myself in the tuberculosis literature with one question in mind: what immunological parameters discriminate the immunity in the large majority of the infected who remain healthy, the “healthy infected”, and in the much smaller number that come down with disease? An answer to this question was necessary for thinking about how both effective vaccination against and immunotherapy of disease might be realised. Although there were tantalising clues, there was no clear, unequivocal answer. This contrasts with many other infectious diseases when the same question is asked. The title of a relatively recent review on the immunology of tuberculosis nicely expresses the dilemma: “In search of a new paradigm for protective immunity to TB”. This paper summarises the evidence for a Th1 response being protective as well as the paradoxes that this “central dogma” leads to, and therefore the need for new insights [1]. Our similar recognition of some of these difficulties was initially unsettling. There was one paradox of which we became acutely aware. It seemed that, when certain immunological parameters are measured, some can discriminate between the immune state of a population of patients from a population of the healthy infected. We illustrate the paradox by considering one such parameter, the level of mycobacterium-specific IgG antibody. This was clearly higher on average in patients than in the healthy infected [2,3]. A paradox became apparent in the context of the assumption that such a parameter should, at least roughly, discriminate the immune state of all patients from those of all the healthy infected, as a truly discriminating parameter would do. However, many patients had considerably lower levels of IgG antibody than the large majority of the healthy infected! This helped us to catch up with others in appreciating the complexities of the immunology of TB. 

As I outline below, some of our observations in the mouse model of human cutaneous leishmaniasis were initially surprising but primed us to consider a radical proposal as to the differences in the nature of immunity in TB patients and the healthy infected. Our basic research on how immune responses are regulated was also critical. The concepts underlying this research may be more difficult for many to consider sympathetically. This is because they are not based on the main and popular frameworks employed over the last three decades in analysing how immune responses are initiated and their Th1/Th2 phenotype determined. I think it helpful if I briefly indicate why I feel these popular frameworks are implausible. I hope this will allow the reader to give a more sympathetic ear to our alternative ideas that are pertinent to our view of the immunology of tuberculosis.

## 2. An Assessment of the Role of Danger-Associated Molecular Patterns (DAMPS)/PAMPS in Initiating Immune Responses and Affecting the Class of Immunity Generated 

The beliefs in the central role of DAMPS/PAMPs in activating CD4 T cells and so initiating immune responses [4,5], and their primary role in determining the class of the ensuing immunity [6,7,8,9], are prevalent. I first summarise why I think these beliefs are implausible and so are an impediment to progress [10,11]. 

A logical consequence of the “DAMP/PAMP-centric” view is that the regulation of the immune response is different for invaders harbouring different PAMPs. This view precludes generalisations concerning how the class of immunity is regulated against diverse pathogens. In contrast, I argue that the evidence supports the plausibility of such generalisations and allows both the development of plausible strategies of vaccination and of treatment [12,13]. Such generalisations also provide the strongest evidence against the “DAMP/PAMP-centric view”. I hope the reader will consider a brief explanation of my views on how the Th1/Th2 phenotype of a response is determined before considering their pertinence to the immunology of tuberculosis. 

## 3. Evidence for Variables of Immunization That Generally Determine the Th1/Th2 Phenotype of the Ensuing Response

I have outlined elsewhere in some detail reasons for my scepticism of the “DAMP/PAMP-centric view” [10,11]. I shall shortly summarise these considerations. However, before doing so, I would like to comment on the overall perspective being developed. Firstly, I argue there are plausible generalisations governing how immune responses are regulated, and that these generalisations can provide a basis for developing general strategies of vaccination and of treatment of diverse infectious diseases. I shall, on this basis, only consider how BCG vaccination might be improved when considering vaccination against tuberculosis. I do not consider second-generation TB vaccines, such as subunit vaccines or vector-based vaccines, that so far also fail to be broadly effective. I suggest the reasons for their failure are similar to those for BCG. If there is a way of making BGC vaccination universally efficacious, the consideration of second-generation vaccines loses its urgency. 

Secondly, I will only consider the prevention and treatment of tuberculosis in “healthy” people. With the AIDS pandemic, and other “unusual” situations, such as malnutrition and vitamin A deficiency [14,15,16,17,18,19,20,21,22,23], resulting or potentially resulting in suppression of immunity as expressed in healthy individuals, it is recognized that some tuberculosis may arise as a result of immunosuppression of an otherwise protective response [14,24,25,26,27]. This adds a degree of complexity to both prevention and treatment. We confine ourselves to situations without such complications. It seems wise to try to develop an understanding of how immune responses are regulated in healthy individuals before attempting to address the real or potential problems associated with immunosuppression. 

The dose of a non-multiplying antigen, or the number of slowly multiplying entities that constitute an infection or invasion, are critical in determining the Th1/Th2 phenotype of the ensuing response. This was first shown by Salvin in the 1950s in terms DTH, as an expression of cell-mediated immunity, and IgG antibody [28]—see Figure 1. Lower doses and numbers favour exclusive Th1, cell-mediated responses, and higher doses and numbers lead to responses with a substantial or predominant Th2 component and the production of IgG antibodies. I stress this generalisation does not apply to responses against rapidly multiplying organisms. Infection with a single, rapidly multiplying bacterium will rapidly constitute a high dose of antigen and result in the generation of Th2 cells and antibody. We illustrate this generalisation, concerning the dependence of the Th1/Th2 phenotype of a response on antigen dose/number of organisms, with a few examples. It holds for foreign, vertebrate, and so PAMP-free antigens, such as proteins [28], foreign, erythrocytes in mice [29,30], for PAMP-expressing mycobacteria in mice [31,32] and cattle [33], and for a protozoan infection in mice [34,35]. This same dependence for PAMP-free antigens and diverse PAMP-expressing entities is consistent with a common, PAMP-independent mechanism, as outlined below.

A given antigen challenge often first results in an exclusive Th1 response that evolves with time to have a significant, and increasingly predominant, Th2 component [28]. This pattern is again seen in responses against diverse foreign, vertebrate and so PAMP-free antigens, and against diverse PAMP-expressing entities, as reviewed elsewhere [11]. This includes responses against HIV-1 [36]. Again, it seems most likely that these common patterns can best be explained by a PAMP-independent mechanism.

A prediction of the mechanism I proposed in 1974, for how the cell-mediated/humoral nature of a response is determined, is that the partial depletion of CD4 T cells, in an animal that would mount a predominant Th2 response in the intact state, would result in a modulation of the immune response towards a Th1 phenotype [37]. We have tested this prediction in many different experimental systems in responses to foreign, vertebrate, and so PAMP-free antigens [38,39,40,41,42]. Parallel observations have been made in the mouse model of the human disease of cutaneous leishmaniasis, caused by the intracellular, protozoan pathogen, *Leishmania major*. Susceptible BALB/c mice, infected with a million parasites, rapidly generate a predominant Th2 response that fails to control parasitaemia. Partial depletion of CD4 T cells around the time of infection deviates the response into a stable Th1 mode and so control of the pathogen [43,44]. In these situations, the only difference between the two groups that result in a predominant Th1 response and one with a substantial Th2 component is the number of CD4 T cells available. This generalisation is paradoxical from the PAMP-centric view, as the number of CD4 T cells is not expected to affect the PAMPs expressed.

## 4. The Threshold Hypothesis: How the Th1/Th2 Phenotype of a Primary Response Is Determined

Central to a consideration of how the Th1/Th2 phenotype of a response is determined are the requirements to activate naïve CD4 T cells. I have argued, and we have provided evidence that, just as the activation of most B cells and CD8 T cells requires activated CD4 T helper cells, so does the activation of naïve CD4 T cells themselves [44,45]. This model contrasts with the popular DAMP/PAMP models. I think it inappropriate to make a substantial case here for the view I favour, though central to the analysis presented. I provide references where this issue has been recently discussed at some length [45,46]. The Threshold Hypothesis is based on the idea that the activation of CD4 T cells requires CD4 T cell/CD4 T cell interactions.

The Threshold Hypothesis was proposed in 1974, in large part because it accounted for the variables of immunization then known to affect the cell-mediated/antibody nature of the ensuing response—see Figure 1. A virtue of this proposal is its quantitative nature. The hypothesis states in its contemporary formulation that tentative and robust interactions between CD4 T cells, mediated by antigen and an antigen-specific B cell acting as an antigen-presenting cell, leads, respectively, to the generation of Th1 and Th2 cells [37,47]. This hypothesis accounts for why minimally foreign antigens [48], such as minor histocompatibility antigens, for which there are relatively few CD4 T cells, induce only cell-mediated immunity. More foreign antigens, for which a greater number of CD4 T cells exist, can generate cell-mediated or IgG antibody responses, depending on the circumstances of immunization—see Figure 1. Immunization with low amounts of antigen supports only weak CD4 T cell cooperation and so tentative interactions and the generation of Th1 cells and cell-mediated immunity. Immunization with medium amounts of antigen supports stronger CD4 T cell collaboration, and the more rapid generation of Th1 cells. Antigen stimulates the CD4 T cells to multiply. Thus, if the antigen level is sustained, the CD4 T cell cooperation becomes stronger and may become robust, leading with time to the generation of Th2 cells and production of IgG antibody. Immunization with optimal amounts of antigen for CD4 T cell cooperation leads more rapidly to robust CD4 T cell cooperation and the generation of Th2 cells; the initial phase of the response, when Th1 cells are exclusively generated, may be transient or even obliterated—see Figure 1. The Threshold Hypothesis also predicted that a partial depletion of CD4 T cells, around the time of antigen impact, will modulate the response from a Th2, humoral to a Th1, cell-mediated phenotype, as is found to be the case in several instances [38,39,40,41,42,43,44]. 

## 5. Beyond Primary Immune Responses

Studies in the 1960s showed immune responses could be locked into a humoral [49] or cell-mediated mode [50], phenomena we respectively refer to as humoral and cell-mediated immune deviation. The mechanism underlying the lock into an antibody mode is most likely responsible for the decrease in the generation of Th1 cells and cell-mediated immunity as Th2 cells are generated and IgG antibody is produced, as occurs upon immunization with a medium or optimal dose of a more foreign antigen—see Figure 1. Most importantly, Parish showed that repetitive exposure of rats over several weeks to low amounts of antigen locked the immune response into a cell-mediated DTH mode [50]. We confirmed in the 1970s the predictions [37] that humoral immune deviation is associated with antigen-specific T cells (CD4 T cells) that inhibit the generation of cell-mediated immunity in the form of DTH [47], and a lock of the response into a cell-mediated DTH mode with the generation of antigen-specific T cells (CD8 T cells) that inhibit IgG antibody responses [47]. 

A further question, central to immunotherapy of disease, is how are on-going immune responses regulated, and so how can they be modulated? Much less research has been performed that bears on this question. We confirmed [51] the conclusion of others [52] that treatment of visceral leishmaniasis patients with anti-parasite drugs modulates the response from one having a substantial Th2 component into a predominant, Th1, DTH mode, similar to the immunity of the healthy infected individuals. Treated individuals resist further infections; their response appears to be locked into a protective, cell-mediated mode. We suggest that such modulation of the response in treated patients occurs due to a decrease in parasite presence upon drug treatment and so of antigen load [51]. We also studied how on-going immune responses are regulated in the mouse model of cutaneous leishmaniasis. BALB/c mice, infected with 3000 *L. major* parasites, express a semi-stable Th1/Th2 response two months post-infection. We modulated this response into an exclusive Th1 mode by partial depletion of CD4 T cells at this time [53]. 

We suggest on the basis of these observations that [54,55] on-going, mixed Th1/Th2 responses can be modulated to a Th1 mode by either decreasing the antigen load, as seen in the treatment of human visceral leishmaniasis [51], or reducing the number of CD4 T cells, as seen in the mouse model of cutaneous leishmaniasis [53]. The evidence for this generalization is less strong than the generalization concerning the role of antigen dose and number of CD4 T cells in affecting the Th1/Th2 phenotype of primary responses. However, the inference concerning the regulation of on-going immune responses is strengthened by being parallel to the conclusion concerning the determinants of the Th1/Th2 phenotype of primary responses. Interestingly, a substantial fraction of HIV-1 infected individuals control virus production after cessation of anti-retroviral treatment, the so-called “post-treatment controllers”. What underlies this phenomenon is generally considered to be enigmatic. We have argued that such “treatment” may have a similar basis as the treatment just outlined for visceral leishmaniasis and that the treatment can be modified to provide generally efficacious and personalized treatment of HIV-1 infections [56]. 

## 6. Vaccination against Pathogens Primarily Susceptible to Cell-Mediated, Th1 Immunity

I suggest the evidence for the importance of the dose of antigen, and the number of slowly growing organisms of an infection, in affecting the Th1/Th2 phenotype of the ensuing response, and its potential significance for medicine, have been surprisingly ignored by the immunological community. Particularly significant in my mind are our own findings [31,32,34,35,55], and those of others [33], that infection with low numbers of slowly multiplying entities can not only induce a sustained Th1 response but a Th1 imprint; challenge with a higher number of the multiplying entities, that in naive animals results in time in a predominant Th2 response and copious IgG antibody production, results in a sustained and predominant Th1 response if the challenge occurs sufficiently long after the first infection. We have shown this “low-dose” vaccination strategy to work in mice for responses to different tumours, as described elsewhere [57]. We also showed the immune response in mice to mycobacteria could be locked into a cell-mediated mode by this strategy [31,32]. We first explored the “low-dose” strategy in the mouse model of the human disease, cutaneous leishmaniasis, caused by the intracellular parasite, *Leishmania major* [34,35]. BALB/c mice are susceptible on the basis that, upon infection with a million parasites, they rapidly generate a Th2 response and are unable to control parasitaemia. We showed that BALB/c mice, on infection with three hundred parasites, generate a stable Th1 response. When challenged about two months later with a million parasites, they again generate a very substantial and stable Th1 response associated with the control of parasitaemia. This is the basis of our low-dose vaccination strategy. It also was known that different strains of mice generate different kinds of immune responses when challenged with a million *L. major* parasites, most strains generating a stable Th1 response, and thus being dubbed resistant [58]. We decided to explore whether the dependence of the Th1/Th2 phenotype of primary immune responses on parasite number is generally true. 

## 7. The Dosage Rule Holds in Diverse Strains of Mice

We injected diverse numbers of the same strain of parasites, by the same route, into mice of diverse strains. We found in general that lower numbers generated a stable Th1 response and higher numbers generated with time responses with a substantial or predominant Th2 component [35]. We could readily define for each strain of mouse a transition number of parasites, N_t_. Infection with a number of parasites below N_t_ generates a stable Th1 response, and with a number above N_t_, a response that, with time, develops a substantial Th2 component. Infection with a number substantially above N_t_ rapidly led to a predominant Th2 response. We found that the value of N_t_ varied for different strains of mice over an approximately million-fold range, being about 500 parasites for the “susceptible” BALB/c strain, and about 500 million parasites for “resistant” CBA mice. This wide range initially staggered us. 

On reflection, we recognized that the genetic diversity underlying this wide range provides us as a species with protection against the unknown. Only about 1% of HIV-1 infected individuals generate a sustained and stable Th1 response: the elite controllers. These must be those rare individuals with an N_t_ above the typical infectious dose of virus. This is a case where, without the intervention of modern medicine, most of those infected would seroconvert and die, but not all, due to our genetic diversity [56]. 

## 8. Protective Immunity against Tuberculosis

As already indicated, the nature of the protective immunity against *M. tuberculosis* is unclear [1]. An appreciation of the conflicting evidence surrounding the importance of Th1 cells in protection provides an appropriate context for considering novel proposals for what immunological parameters might distinguish the immunity of the healthy infected and patients.

Infection of humans or animals with *M. tuberculosis* results in IFN-γ-producing Th1 cells [59,60]. Given that the large majority of people infected with the pathogen do not develop disease, it was natural to explore the role of such cells in protection in animal models of TB. Deficiency in the ability of mice to produce IFN-γ leads to extreme susceptibility [61,62]. Moreover, given that the pathogen infects macrophages, the biological functions of IFN-γ in activating anti-microbial pathways in macrophages support the idea that IFN-γ was central [63,64,65,66,67,68,69,70,71]. The pertinence of these observations to human tuberculosis was supported by genetic deficiencies in humans leading to extreme susceptibility to mycobacterial infection [72]. However, deficiencies in pathways other than those regulated by IFN-γ lead to susceptibility [73,74,75,76,77,78,79]. These observations appear to support the idea that Th1 cells are protective and that IFN-γ production alone is insufficient for protection. 

Another approach used was to examine the immunological parameters associated with different protocols of vaccination and assess their correlation with protection [80]. Protection did not correlate with degree of IFN-γ production on vaccination. However, the logic underlying this strategy is questionable. It is important to assess the immunological state associated with the challenge rather than with priming when attempting to determine the immunological parameters distinguishing protection and lack of protection. Another approach was to examine the immunological parameters associated with a primary infection that sometimes did or sometimes did not lead to disease. Observations in people, monkeys, and mice did not show that greater production of IFN-γ was always associated with resistance but was sometimes associated with disease [81,82,83]. This was taken to mean that Th1 cells producing IFN-γ were not the critical cells in providing protection. I explain later why I think there are limitations to this argument. 

## 9. Exploiting the Relative Prevalence of IgG Subclasses among Antigen-Specific Antibody to Infer the Th1/Th2 Phenotype of the Response

The methodology underlying our attempt to find correlates distinguishing the immunity of the healthy infected and of TB patients was based on our studies in mice. We had, in many studies, directly assessed the Th1/Th2 phenotype of immune responses in mice infected with *L. major* by sacrificing the mice and assessing the prevalence of parasite-specific cytokine-producing Th1 and Th2 cells in their spleen. We also assessed the levels of parasite-specific IgG_2a_ and IgG_1_ antibodies. The correlation of either no IgG antibody, or prevalence of IgG_2a_ antibody with predominant Th1 responses, and of predominant IgG_1_ antibody with predominant Th2 responses, was striking. As we were interested in longitudinally monitoring immune responses following infection, we started using the relative prevalence of IgG_1_ and IgG_2a_ antibodies among parasite-specific antibodies, or the IgG_1_/IgG_2a_ ratio, as an indirect measure of the Th1/Th2 phenotype of on-going immune responses: low and high ratios corresponding, respectively, to predominant Th1 and Th2 responses. This “IgG_1_/IgG_2a_ methodology” was very useful in our studies in mice on immune responses to *L. major* [34], mycobacteria [31,32], and tumours [57]. We decided to examine the levels of the different IgG subclasses among mycobacterium-specific antibodies of the healthy infected and TB patients. Our “healthy infected” were healthy healthcare workers looking after TB patients who were PPD positive but showed no clinical symptoms. Although we assessed the levels of all IgG subclasses, we found the most illuminating parameter to be the IgG_1_/IgG_2_ ratio [84].

## 10. The Hypothesis of Type 1 and Type 2 Tuberculosis: Two Distinct Types of Failure by the Immune System to Control the Pathogen

The IgG_1_/IgG_2_ ratio of the mycobacterium-specific antibodies of the healthy infected covered a thousand-fold range, from 0.001 to 1. The ratio among patients covered a hundred, thousand-fold range, from 0.001 to 100. We anticipated from the literature that the higher ratios reflected a more substantial Th2 component of the response. It was surely natural to think that those patients with a higher IgG_1_/IgG_2_ ratio than those found among the healthy infected had an inadequate Th1 protective immune response due to its downregulation associated with the generation of a substantial Th2 component. The immune response of such individuals would be analogous to those of visceral leishmaniasis patients. We referred to this form of the disease as type 2 tuberculosis. The other TB patients had an IgG_1_/IgG_2_ ratio in the same range as that of the healthy infected, presumably associated with a predominant Th1 response. We refer to such patients as having type 1 tuberculosis. Our observations immediately explained why no parameters could discriminate between the immunity of all patients and the immunity of all the healthy infected [84]. It naturally led to the question of how could a “protective response” lead to the failure of immunity to contain the pathogen? 

## 11. Ideas on How Th1, “Protective Immunity”, May Be Associated with Failure to Control the Pathogen

Having immersed myself in the paradoxes of the literature on the immunology of TB, as well as having worked in the experimental model of human cutaneous leishmaniasis, I found myself with a ready hypothesis for how a predominant Th1 could be associated with failure to control the pathogen.

Conventional vaccination works against pathogens susceptible to antibodies because vaccination results in a more rapid and greater antibody response. This understanding brings out the importance of the speed of an effective immune response in providing protection against a multiplying invader. Conventional vaccination provides the host with the advantage of establishing effective immunity and the unrestrained multiplication of the pathogen that leads to disease. Similarly, it may be that a “protective Th1 response” is generated but is sometimes of insufficient size to contain the pathogen [13].I was impressed that mice of some strains, “resistant” to *L. major* and infected in the foot with the standard number of a million parasites, had a grossly swollen foot over weeks, but the lesion then resolved [34,35,53]. The swelling was surely due in part to the Th1 inflammatory response against the considerable burden of parasites in the foot. This increase in foot width has been used for years as a rough and ready measure of disease severity. However, by this criterion, the “disease” in resistant mice resolved! This resolution was why the mice were described as resistant. However, I was impressed by this prolonged and substantial lesion. It seemed obvious to me that the lesion, if it occurred elsewhere in the body, such as the lung, might cause considerably greater distress and pathology. These mice generated a sustained Th1 response during the course of the formation of the lesion and its resolution. It therefore seemed that it took a considerable time for these mice to generate a sufficiently strong immune response to kill parasites more rapidly than they were increasing through multiplication. Once this happened, the lesion presumably resolved.We recognized that “susceptible” BALB/c mice had the intrinsic genetic capacity to mount protective Th1 responses. They mounted such a response when infected with three hundred rather than a million parasites, or when infected with a million parasites and their CD4 T cells were substantially depleted. These parasites were more “immunogenic” in BALB/c than in “resistant” mice. It seems that the large range in the value of N_t_ for different mouse strains must reflect genetic polymorphisms that greatly affect the immunogenicity of the parasites. The parasites were most immunogenic in BALB/c mice that had the lowest value of N_t_. It is recognized that the more immunogenic an antigen is, the lower is the dose able to generate an immune response. Consider a pathogen only susceptible to cell-mediated attack and the responses to an infection of individuals with vastly different N_t_s. When the infective dose, N_i_, is above N_t_, an ineffective Th2 response will in time be generated, associated with the down-regulation of the protective Th1 response and so with disease. If N_i_ is below N_t_, a stable and predominant Th1 response will be generated. In all cases, there will be a lag period before there is a substantial Th1 response. If this lag period is short, as expected if N_i_ is close to but below the value of N_t_, immunity will be induced before the pathogen burden has greatly increased, and so a relatively small protective immune response should be able to contain the pathogen. This would likely reflect the situation in a healthy infected individual. Consider the case where there is a much longer lag period, as might be expected of an individual with a very high N_t_, much higher than N_i_. In this case, the pathogen burden is likely to greatly increase before significant immunity begins to be established. Moreover, given that the pathogen burden is now larger, a larger immune response would be required to turn the tide. The response may not even reach such an intensity. If it does, so the immune response now kills the pathogen at a greater rate than the pathogen is increasing by multiplication, and a spontaneous cure may occur. If not, chronic or progressive disease would presumably ensue.

## 12. Two Types of Tuberculosis

We encapsulate the ideas outlined above in the form of a hypothesis. There are diverse host genes that affect immune responses to *M. tuberculosis*, not to mention variants of the pathogen. Employing the mouse model of cutaneous leishmaniasis, we studied how host genes play a role in resistance/susceptibility through their effect on the nature and size of the immune response against the pathogen. This pathogen can only be contained by a sufficiently strong, Th1, cell-mediated attack. We did not attempt to analyse the basis of this genetic diversity in terms of host genes, but rather the effect of these genes on the host’s resistance or susceptibility to the pathogen. We also worked within a justified framework that there are valid generalisations concerning how certain variables of immunization/infection affect the nature of the ensuing response. Central to our considerations were the generalisations that the number of multiplying entities constituting the infection, and the time after the infection, were critical, as these variables seem to affect the Th1/Th2 phenotype of the response to many antigens in general [10,11] and to mycobacteria in particular [31,32,33]—see Figure 1.

Our hypothesis proposes that there are four possible fates upon infection by *M. tuberculosis* in the absence of medical treatment. The healthy infected make a sufficiently strong, stable, predominant Th1 response relatively early after infection, and so can contain the pathogen at low levels with a modest response. These individuals are infected with a number of mycobacteria, N_i_, that is below their N_t_ for the particular strain of pathogen involved, and where N_i_ and N_t_ do not differ by orders of magnitude. Individuals whose N_t_ is lower than N_i_ generate a rapid immune response that evolves with time to have a significant or predominant Th2 component, resulting in down-regulation of the protective Th1 response, lack of containment of the pathogen and type 2 TB. Miliary tuberculosis [85], rapidly fatal if not treated, is probably an extreme form of type 2 tuberculosis, where the response has evolved into a predominant Th2 mode. 

Individuals infected with N_i_ organisms that have an N_t_ orders of magnitude greater than N_i_ are envisaged to be unable to make an immune response shortly after infection, as the level of the “antigen” is too low. Their immune system only starts making a significant immune response when the pathogen has multiplied considerably. Because of this increased pathogen burden, a much larger protective response is required to contain the pathogen. Such a large response may never be achieved, or only a long time after infection. We refer to these individuals as having type 1 tuberculosis. If the immune response eventually reaches a size where there is a net decrease in pathogen burden, there will presumably be a spontaneous cure, if irreversible pathological damage has not been inflicted. (This is unlikely for most patients as antibiotic therapy of tuberculosis, if properly carried out, results in effective treatment, from which we infer that irreversible damage has not been caused.) It is very interesting to note here that some patients, before the advent of antibiotics, are known to have spontaneously been cured [86]. It seemed very difficult to envisage how this occurred if all patients were ill because their response had a significant and detrimental Th2 component, as responses are not known to “go backwards” under natural conditions from a mixed Th1/Th2 to a Th1 mode [54]. If the immune response does not reach sufficient intensity, or irreversible damage has been inflicted, we would have chronic/progressive disease. 

## 13. Paradoxes

Three paradoxes have been much discussed in the context of the idea that Th1 responses are protective against *M. tuberculosis.* I would like to consider how these three paradoxes can be resolved by our hypothesis, which postulates a sufficient Th1 response is protective. Firstly, our hypothesis has quantitative features and so brings quantitative considerations to the fore. Such quantitative considerations are essential to resolving some of the paradoxes. The first paradox is the lack of immunological parameters clearly discriminating between the immunity of the healthy infected and of patients. This paradox is resolved by the proposal that the immune system can fail to contain the pathogen for two different reasons. The lack of discriminating parameters is strikingly illustrated by the range of the IgG_1_/IgG_2_ ratio we found among patients and the healthy infected. Those patients with a higher ratio than the ratio seen in the healthy infected obviously have a different kind of immunity from that of the healthy infected. Those patients with a similar ratio as the healthy infected have, as do the healthy infected, a predominant Th1 response. We suggest that this response in type 1 TB patients is inadequate to contain the multiplying pathogen, and so chronic or progressive disease ensues [84]. 

Secondly, a long-standing paradox is that Th1 immunity is responsible both for protection as well as primarily responsible for granuloma formation in the lung and so for the pathology of the most common form of tuberculosis [1]. The paradox is, how do these two types of immunity differ? How can a particular form of immunity be both protective and causing pathology, or are they subtly different? We suggest, based on our hypothesis, that they do not necessarily differ. The same kind of response can provide protection and be responsible for pathology when quantitative considerations are brought into play [55,84], as we hope to have made clear above. 

Lastly, a study was undertaken to longitudinally examine the nature of the immunity in individuals recently infected by *M. tuberculosis*, before it became apparent which individuals would remain healthy and which would develop disease. It was assumed that the immunity of those who contained the pathogen and so remained healthy could be differentiated from the immunity of those who became ill, thus indicating the type of immunity that is protective. It appears likely that type 1 tuberculosis is considerably more prevalent than type 2, and so I shall, for simplicity and for this analysis, assume the individuals who became ill had type 1 tuberculosis. 

The hypothesis I am putting forward predicts that those infected individuals who will in the long term be characterized as a healthy infected individual, will, relatively shortly after infection, make a protective Th1 response that can relatively rapidly contain the infection. These individuals will not need a very large response to contain the modest bacterial loads. Indeed, if the response is sufficiently large that it kills the pathogen faster than the pathogen replicates, the number of pathogenic organisms will decrease, the antigen load will decrease, and the intensity of the immunity will decline. Thus, a steady state is reached, the size of the response reflecting the pathogen load.

As outlined above, we expect an individual with an exceptionally high N_t_ will initially not generate a significant immune response if N_i_ is much smaller than the individual’s N_t_. The infected individual with a N_i_ closer to but lower than its N_t_, destined to remain healthy, will initially more rapidly generate a stronger Th1 response than the individual destined to develop type 1 tuberculosis. However, the bacterial burden will increase over a prolonged period in the latter individual before significant protective immunity is induced. As the immunity is insufficient to contain the bacterial burden, the generation of immunity, once initiated, increases in intensity, as the bacterial burden and so antigen load continues to increase. Thus, if we examine the immunity some time after infection, we would expect those individuals destined to develop type 1 tuberculosis would have a greater Th1 response than those destined to remain healthy. The study showed that those destined to develop tuberculosis had, at some time after infection, stronger Th1 responses than those destined to remain healthy [80]. This was considered to be paradoxical on the idea that Th1 responses are protective, but this observation is expected on our hypothesis. 

## 14. Vaccination and Treatment

I have discussed my ideas on these subjects elsewhere [12,13], and so I will only summarize my suggestions.

Vaccination against TB may be interfered with by exposure to environmental mycobacteria, causing imprints upon the immune system. We therefore suggest vaccination should be carried out neonatally, as already widely practised [12].

We base our proposals on the idea that infection with low numbers of BCG (<N_t_) induces stable Th1 responses and Th1 imprints and will protect against both type 1 and type 2 tuberculosis. We suggest inoculation with a number of BCG organisms that is lower than the N_t_ for all individuals of the population will either be immunogenic and induce a Th1 imprint in some individuals, or be below the immunogenic threshold to immediately induce immunity. In the latter case, the BCG will multiply until it reaches the immunogenic threshold, when it induces a chronic Th1 response and a Th1 imprint. Thus, inoculation with a sufficiently low dose of BCG may provide universal protection [13]. Two facts are of interest in this context. Buddle tested the low-dose strategy in cattle against experimental tuberculosis. It had been found previously that BCG vaccination did not produce reliable protection in cattle. Buddle and colleagues reduced the number of BCG employed for vaccination by about a million-fold and achieved dramatic protection [33]. We examined the possibility of generating Th1 imprints in very young mice with BCG. Many refer in the literature on the mouse model of TB to a million BCG as a low number. We found the best Th1 imprints were achieved by injecting 20 “colony-forming units” of BCG into the very young mice, the lowest number we employed [32]. 

That this vaccination strategy will provide protection against type 2 tuberculosis is supported by Buddle’s vaccination study in cattle, and our studies in mice with mycobacteria and *L. major*. We also suggest it will provide protection against type 1 tuberculosis. The basis of immunological memory was envisaged by the formulators of the Clonal Selection Theory to be, in part, the result of the increased frequency of antigen-specific lymphocytes. We think this low-dose vaccination strategy will result in memory Th1 responses, more rapid and greater than primary responses, and so will protect individuals with a high N_t_, who, if not vaccinated, would develop type 1 tuberculosis.

Koch tried to develop a way of treating tuberculosis [87,88,89,90], as I have recently reviewed elsewhere [55]. I think it is realistic in terms of current knowledge to propose exploratory treatments for TB.

The two types of tuberculosis envisaged here reflect two types of failure by the immune system, and so require different treatments. The first step would be to assess which type of tuberculosis the patient suffered from by measuring their IgG_1_/IgG_2_ ratio. I suggest type 2 tuberculosis can be treated in a similar manner as cutaneous leishmaniasis. Patients would be given standard antibiotic drugs to kill the parasite and so decrease the antigen load. The effects of this on the Th1/Th2 phenotype of their immune response could be longitudinally monitored by following changes in the IgG_1_/IgG_2_ ratio. When this ratio is very small, indicating a predominant Th1 response, treatment would be stopped. We anticipate that the predominant Th1 response would now control and further reduce the pathogen load. A considerable advantage of this treatment over current therapy is that it is personalised and may require only a few weeks. The standard, “non-monitored” treatment of visceral leishmaniasis patients lasts only three weeks. 

The treatment of type 1 tuberculosis that I propose really follows in the footsteps of Koch’s vision. Koch treated TB patients by immunizing them with PPD. He hoped in this manner to stimulate the protective immune response and so cure the patient. This procedure had diverse results, with some patients judged to improve, and others taking a considerable turn for the worse [87,88,89,90]. I describe elsewhere how these different outcomes can be readily understood in terms of our hypothesis of there being two types of tuberculosis [84]. Briefly, the administration of antigen to a patient with type 2 tuberculosis is expected to push the response towards the Th2 pole, and so towards miliary TB. It would exacerbate disease. I discuss the beneficial effects of the treatment below.

We envisage that the size of the protective Th1 in those patients with type 1 tuberculosis is limited by the “low amount” of antigen present. In this case, current antibiotic treatment, from an immunotherapeutic point of view, undermines the insufficiently protective immunity that already exists. We suggest that benign antigen, in the form of BCG, be given together with antibiotic therapy, to maintain and increase the Th1 immunity, as essentially envisaged by Koch. It would be important to monitor the IgG_1_/IgG_2_ ratio to ensure the response did not deviate towards a Th2 mode. If it was, further antigen should not be administered, or the amount reduced. Optimization of this treatment would likely allow much shorter treatment than current antibiotic therapy [55]. 

## 15. The Genetic Diversity of Host and Pathogen

As is evident, the main focus of our research has been to develop strategies of vaccination and immunotherapy that work independently of the host genotype. I believe the proposed strategies are based on valid generalisations as to how immune responses are regulated, and so are applicable to responses against diverse types of pathogens and against tumours. I anticipate they will also be effective against diverse variants of the TB pathogen. The relationship between a particular variant of the pathogen and a particular host must surely be critical in determining the value of the transition number, N_t_, for the particular combination. Indeed, some of the host genetic loci known to affect susceptibility to tuberculosis “make sense” in terms of the general concepts developed here. Thus, the Nramp locus contains different alleles that affect the ability of various intracellular parasites that reside in the phagolysosomal compartment of macrophages, including *M. tuberculosis,* to multiply [91], thereby affecting the antigenic load, and therefore the nature of the immune response. It is known that loci encoding class II MHC antigens affect the individual’s susceptibility to tuberculosis [92]. Originally, immune response genes (Ir-1 genes) were discovered employing very simple antigens, such as poly-L-lysine (PLL). Responders produced antibodies upon immunization, whereas non-responders did not. With time, it became apparent that class II MHC molecules were responsible for mediating Ir-1 gene effects by determining whether the host’s class II MHC molecules could present antigen. In the case of PLL, for example, the class II MHC molecules of responder animals could bind oligo-lysine, and so “responder” APC could present PLL and so T helper cells specific for PLL could be induced; the class II MHC molecules of non-responders could not present PLL and so PLL-specific helper T cells were not activated [13]. It also became clear with time that class II MHC antigens did not only control responses to very simple antigens, such as PLL. “Ir-1 genes” also controlled responses to complex antigens if limiting doses of antigen were chosen for immunization, which could induce antibodies in some strains of mice, for example, but not in other strains. Thus, it became plausible with time that the class II MHC molecules an individual expresses determine the peptide/MHC class II complexes of an antigen that are presented by the individual’s APC, and so the repertoire of CD4 T cells specific for the antigen. Differences in the class II MHC molecules present are reflected in the differences in numbers of CD4 T cells present, and so differences in the Th1/Th2 phenotype of the response under given circumstances, and so differences in resistance/susceptibility to tuberculosis. 

## Figures and Tables

**Figure 1 ijms-24-01887-f001:**
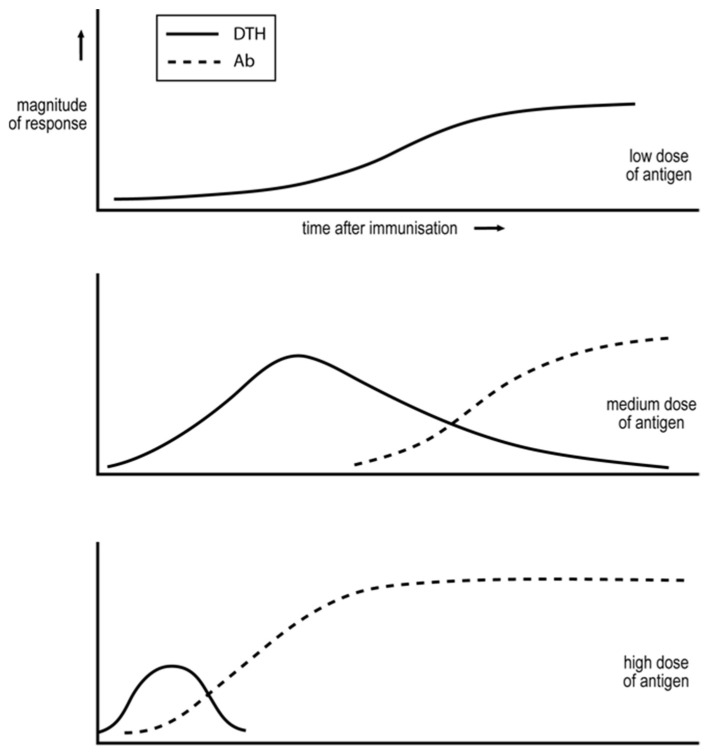
Encapsulation of Salvin’s findings [14].

## Data Availability

Not applicable.

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
