# Peer review of "The Problem of Host and Pathogen Genetic Variability for Developing Strategies of Universally Efficacious Vaccination against and Personalised Immunotherapy of Tuberculosis: Potential Solutions?"

_ijms, 2023, doi:10.3390/ijms24031887_

Round 1
Reviewer 1 Report
This is an interesting opinion article on the mechanisms of immunity against M. tuberculosis. The article is well written and is extremely informative and instructive. I would like to read some more information about the importance of host gene factors and their interaction with the pathogen since it is clear that the course of infection is different between various individuals and this could be due to a different host genetic background affecting immune responses. Also the author could add one or two figures to illustrate his points clearly and to make the article more attractive.
Author Response
I thank the reviewer for their comments and advice. I have in response included a Figure to illustrate the generalization of how dose of antigen and time after antigen impact affect the cell-mediated./IgG antibody nature of the response. In addition, I have expanded the last section to illustrate how two genetic loci of the host likely contribute to the transition number of the host: the nramp locus that controls the rate of multiplication of diverse pathogens that reside in the phagolysosomal compartment of macrophages, and so the "antigen dose", and host MHC class II molecules, that likely affect the Th1/Th2 phenotype of the response by affecting the number of CD4 T cells present in the host specific for a particular pathogen. Addition is in last section and is in red print.
Reviewer 2 Report
The article "The problem of host and pathogen genetic variability for developing strategies of universally efficacious vaccination against and personalised immunotherapy of tuberculosis: potential solutions?" gives a perspective on tuberculosis vaccines according to a series of expertiment, mostly in mice.
This letter takes in consideration the immune response of lymphocytes T1 and T2 helper and their reaction to a certain bacterial load. Also, the author gives his opinion on BCG based vaccine, why it is not effective and some hints to develop revisited and more effective strategies of vaccination.
The article is well written and fully understandable. Anyway, the lack of explanations about the immology part, that are given for granted, may require some integration from the reader.
In my opinion the perspective is limited to the first generation vaccines and doesn't take into account the last generation such as the protein-subunit vaccine or the viral-vectored ones. Those vaccine, exactly like the BCG-based vaccine, gives only a partial coverage. For a complete perspective the other vaccines should be taken into account, too.
This article should ungergo major revision or, maybe, should be published on a journal with a lower impact factor than the IJMS.
Author Response
I thank the reviewer for the comments and thoughts. Some of the comments appear positive, some others less so. Naturally, I have concentrated on the least positive in order to improve the manuscript. I thought the least positive comments are: "In my opinion the perspective is limited to the first generation vaccines and doesn't take into account the last generation such as the protein-subunit vaccine or the viral-vectored ones. Those vaccine, exactly like the BCG-based vaccine, gives only a partial coverage. For a complete perspective the other vaccines should be taken into account, too."
My response here to this comment is two fold: the article surely was not limited to BCG vaccination, as implied, but tried to make the case that there are plausible immunological generalizations that are relevant to diverse situations, including vaccination/treatment of tuberculosis, of cancers and other infectious diseases caused by viruses and protozoa. The focus is on the possibility of valid generalizations and their use in formulating general strategies of vaccination and treatment. Secondly, I had hoped a reader might realize that I was using the failure of BCG to provide universally effective protection to illustrate general issues. I was surprised at the reviewer's asking for detailed considerations concerning other types of vaccine being explored. I had hoped a reader would appreciate the general nature of the considerations developed, and so not request a consideration of every distinct type of vaccine. I have modified the introduction part of the manuscript to add a part (in red script) to explain the generality of the approach developed in the manuscript.
I also have read this reviewer's comments more than once. I am surprised that at the end the reviewer suggests a major revision would be appropriate, or submission to a lower impact journal. I could not decipher from the comments made how this recommendation was justified, except for the above quoted opinion, that made me wonder whether the reviewer had carefully read the manuscript. This review did not provide me with comments that would help me to significantly improve the manuscript. I felt there was little evidence of an attempt to assess the significance of the proposals made, in contrast to another review. I think the aim of the manuscript is clear and ambitious. I would hope reviewers would generally take into account the potential significance of a manuscript when reviewing it, assessing its worthiness and when making recommendations. The reviewer feels "The conclusions are supported by the results". I can only infer that this means the reviewer feels the proposals for universally efficacious vaccination against and treatment of tuberculosis are considered plausible.
Round 2
Reviewer 2 Report
Dear author, thanks for the changes, your paper is more clear and convincing now. Overall, the addiction on point 3 of the paper is more elucidating.
I'm sorry that you think that I didn't read the article with attention, my comments should give you an idea of how carefully I read it, instead. There are still some little oversight, like the lack of a space after the "(78)." citation.
The article is improved and can now be published on IJMS
Author Response
The reviewer felt the revised manuscript was acceptable for publication. I thank the reviewer for his comments and help.